# The Role of Endoscopic Transpapillary Stenting of the Main Pancreatic Duct during the Endoscopic Treatment of Pancreatic Fluid Collections

**DOI:** 10.3390/jcm10040761

**Published:** 2021-02-14

**Authors:** Mateusz Jagielski, Marek Jackowski

**Affiliations:** Department of General, Gastroenterological and Oncological Surgery, Collegium Medicum Nicolaus Copernicus University, 87-100 Torun, Poland; jackowscy@hotmail.com

**Keywords:** transpapillary drainage, transmural drainage, pancreatic fluid collections, acute pancreatitis, endoscopy

## Abstract

Endotherapy is a recognized, widely available, and minimally invasive treatment method for pancreatic fluid collections (PFCs) formed in the course of acute pancreatitis (AP). The use of endoscopic techniques in the treatment of main pancreatic duct (MPD) disruption due to AP remains unclear. In this article, a comprehensive review of current literature referencing our observations was performed to identify publications on the role of MPD stenting in patients undergoing endoscopic drainage of PFCs resulting from AP. In this paper, we attempt to clarify this most controversial aspect of endotherapy for PFCs based on existing knowledge and our own experience regarding the endoscopic treatment of AP sequelae. Endoscopic retrograde pancreatography should be performed in all patients undergoing endoscopic drainage of walled-off pancreatic necrosis to assess the integrity of the main pancreatic duct and to implement endotherapy if pancreatic duct disruption is detected. Passive transpapillary drainage is an effective method for treating MPD disruption in the course of necrotizing AP and is one of the key components of endoscopic therapy for local pancreatic necrosis. Conversely, in patients with pancreatic pseudocysts, passive transpapillary drainage reduces the effectiveness of endoscopic treatment and should not be used even in cases of MPD disruption during transmural drainage of pancreatic pseudocysts. In conclusion, the use of transpapillary drainage should depend on the type of the PFC. This conclusion is of great clinical importance, as it can help improve the results of pancreatic endotherapy for fluid collections resulting from AP.

## 1. Introduction

Acute pancreatitis is associated with high morbidity and mortality, especially if an infection of the necrotic region is present. According to the 2012 revision of the Atlanta Classification of Acute Pancreatitis, depending on the stage of disease and morphological type of acute pancreatitis, four types of fluid collections formed in the pancreatic and peripancreatic regions in the course of acute pancreatitis can be distinguished: acute peripancreatic fluid collection, pancreatic pseudocysts, acute necrotic collection, and walled-off pancreatic necrosis [1,2,3].

In the early stage of the interstitial-edematous form of acute pancreatitis, acute peripheral fluid collections may develop, which, after about 4 weeks, develop into pancreatic pseudocysts [1,2,3]. Acute peripancreatic fluid collection and pancreatic pseudocysts do not contain solid tissue elements and are filled with serous content with high enzymatic activity (amylase and lipase) [1,2]. Most acute peripancreatic fluid collections regress spontaneously without intervention [4,5]. Pancreatic pseudocysts, which are late-phase collections, have well-defined walls, which become thicker and better developed as collections persists, that is, over time from the onset of the acute pancreatitis [1,2,3]. Moreover, pancreatic pseudocysts are also present in patients with complicated chronic pancreatitis [1,2,3].

During the first 4 weeks of acute necrotizing pancreatitis, pancreatic fluid collections are referred to as acute necrotic collections, which are poorly demarcated fluid reservoirs containing a large amount of necrotic tissue that form in most patients over these first 4 weeks of illness [1,2,3]. Nearly half of the acute necrotic collections undergoes spontaneous regression [4,5]. The other half evolves into walled-off pancreatic necrosis, which is a well-demarcated collection of pancreatic fluid forming after the fourth week of the disease course and contains liquified necrosis and fragments of necrotic tissue. The amount of this necrotic tissue depends on the degree of liquification from the necrotic tissue, which in turn depends on the time elapsed since the onset of the disease [1,2,3,4,5].

Over the past decades, major changes have been observed regarding the treatment strategy of pancreatic and peripancreatic fluid collections in the course of acute pancreatitis [6,7,8,9]. It is now thought that asymptomatic pancreatic fluid collections should be treated conservatively. Most post-inflammatory pancreatic fluid collections regress spontaneously without intervention [4,5,6]. However, some patients exhibit clinical symptoms associated with the presence of pancreatic fluid collections, and interventional treatment is necessary [4,5]. The main indication for interventional treatment of the consequences of acute pancreatitis is infected pancreatic and peripancreatic fluid collections [6]. Interventional treatment is also required in patients with clinical symptoms directly associated with the collections, such as compression symptoms (mechanical jaundice, ileus, etc.) [6]. Interventional treatment of post-inflammatory pancreatic fluid collections should take place at least four weeks from the onset of acute pancreatitis [6]. This intervention timing is better due to the encapsulation of the collection [1,2,3,4,5,6]. Patients with asymptomatic pancreatic fluid collections, regardless of size, do not require intervention [6].

In recent decades, minimally invasive techniques for treating the sequelae of acute pancreatitis have dynamically developed [6,9]. Endotherapy is an effective and safe method for the treatment of pancreatic fluid collections, constituting an alternative to other minimally invasive treatment techniques [6,9,10].

The disruption of the main pancreatic duct may occur in the course of acute pancreatitis, manifesting as a leakage of contrast into the reservoir during endoscopic retrograde pancreatography [6,10,11]. Partial disruption to the pancreatic duct presents as a leakage of contrast with the contrast still visible in the distal part of the duct relative to the site of disruption [6,10,12]. Complete disruption is visible as leakage of contrast outside the duct without a contrast of its distal part [6,10,13]. A separate issue is disconnected duct syndrome (pancreatic fragmentation), diagnosed in patients with complete pancreatic duct disruption, or a contrast-filled segment of the main pancreatic duct without contrast flow outside the duct in retrograde pancreatography along with the presence of a distal fragment of the pancreatic parenchyma in other imaging examinations.

Endoscopic treatment of pancreatic duct disruption consists of endoscopic sphincterotomy and stenting of the main pancreatic duct (passive transpapillary drainage) to ensure physiological outflow of pancreatic juices into the duodenum [6,10,11,12,13]. Endoscopic retrograde pancreatography in patients with pancreatic fluid collections resulting from acute pancreatitis is often very challenging due to failed pancreatic duct cannulation caused by duodenal deformation or impression, which prevents the introduction of the device (luminal compression), or the impossibility of identifying the major duodenal papilla in the duodenal inflammatory infiltration [10]. While endotherapy is a widespread method used for treating pancreatic post-inflammatory fluid collections, the use of endoscopic techniques in the treatment of main pancreatic duct disruption in the course of acute pancreatitis remains unclear. Currently, there are no guidelines clearly defining the role of main pancreatic duct stenting during endoscopic drainage for post-inflammatory pancreatic and peripancreatic fluid collections.

Few publications are available in the current literature about passive transpapillary drainage in patients with acute pancreatitis, and the available data are often contradictory. In this paper, a comprehensive review of current literature referencing our observations from pancreatic referral centers was performed to identify publications on the role of pancreatic duct stenting in patients undergoing the endoscopic drainage of pancreatic fluid collections resulting from acute pancreatitis. In this publication, we attempted to clarify this most controversial aspect of endotherapy for pancreatic fluid collections based on existing knowledge of the endoscopic treatment of acute pancreatitis sequelae. Due to the our substantial experience in the field of endotherapy of the consequences of acute pancreatitis, and because few publications were available, this review article is, to a large extent, based on our own observations derived from data available in the current literature. Consequently, we tried to avoid conclusions drawn from the publications and rather speculate on the mechanisms and requirements for future study in the context of a randomized series.

## 2. Clinical and Research Consequences

Endotherapy for post-inflammatory pancreatic and peripancreatic fluid collections (Figure 1A–L) is a recognized, minimally invasive treatment method [6,7,8]. Transpapillary endoscopic drainage involves accessing the collection through the major duodenal papilla if the main pancreatic duct communicates with the collection [6,10,11,12,13]. Active transpapillary drainage involves the introduction of a nasal drain and a pancreatic stent through the major duodenal papilla with their distal ends passing through the site of disruption into the lumen for the fluid collection [6,10,11,12,13]. Subsequently, the collection is rinsed with a saline solution through the drain. Passive transpapillary drainage involves the introduction of a stent into the main pancreatic duct, which is important for endotherapy for main pancreatic duct disruption [6,10,11,12,13]. The size and length of the pancreatic stent should be selected individually according to the fluoroscopic image of the main pancreatic duct during endoscopic retrograde pancreatography. Main pancreatic duct stenting (passive transpapillary drainage) is designed to ensure the free outflow of pancreatic juices by physiological means into the duodenal lumen and, consequently, to prevent the juices from escaping through the injured duct into the collection, thus increasing its volume. Moreover, inserting a stent into the pancreatic duct (Figure 2A–C) is supposed to create conditions that promote healing and stop pancreatic juice leakage by bridging the disrupted part of the duct.

### 2.1. Walled-Off Pancreatic Necrosis

A 2018 publication presented the results of the endoscopic treatment of 226 patients with walled-off pancreatic necrosis [10]. Disruption of the main pancreatic duct in acute necrotic pancreatitis was present in 166 (81.37%) patients [10]. It was demonstrated that patients who had not undergone endoscopic retrograde pancreatography during endoscopic drainage had significantly worse outcomes after endoscopic treatment with a lower efficacy of the endotherapy for pancreatic necrosis, a greater number of recurrent pancreatic fluid collections, and worse long-term effects of treatment for the walled-off pancreatic necrosis, compared to patients who had undergone endoscopic retrograde pancreatography [10]. Thus, the utility of endoscopic pancreatography in patients with pancreatic necrosis for assessing the integrity of the pancreatic duct was confirmed. Most patients with pancreatic necrosis will be diagnosed with duct disruption during pancreatography, and it will be necessary to use endoscopic techniques to treat the disruption of the main pancreatic duct. Stenting of the main pancreatic duct during transmural drainage in patients with walled-off pancreatic necrosis will improve the results of endoscopic treatment.

There is no clearly defined timing for passive endoscopic transpapillary drainage. The basis of the treatment is to perform passive transpapillary drainage during ongoing transmural drainage of walled-off pancreatic necrosis—never in reverse. Transpapillary drainage performed before transmural drainage leads to secondary infection of the necrotic collection. Consequently, it worsens the clinical condition of the patient due to the inability to perform complete drainage via the transpapillary route. Consequently, active transmural drainage of pancreatic necrosis with endoscopic necrosectomy should be performed first, and only during ongoing transmural drainage should the endoscopic pancreatic duct stenting be performed.

### 2.2. Pancreatic Pseudocyst

Two other studies presented the results of endoscopic treatment of pancreatic fluid collections, where the vast majority (93–100%) of the study group consisted of patients with pancreatic pseudocysts [14,15]. Both publications reported completely contradictory results [14,15] compared to those presented above. They demonstrated that pancreatic duct stenting did not improve the results of the endoscopic treatment of patients undergoing transmural pancreatic pseudocyst drainage and adversely affected the long-term effects of endotherapy for pancreatic fluid collections [14,15]. Thus, both publications corroborated our hypothesis that in patients with pancreatic pseudocysts, passive transpapillary drainage should not be performed during the transmural drainage, even in cases of confirmed main pancreatic duct disruption. Stenting of the main pancreatic duct during transmural pancreatic pseudocyst drainage reduces the effectiveness of endotherapy.

Moreover, patients with pancreatic pseudocysts and pancreatic duct disruption should undergo transpapillary drainage or transmural drainage of the pseudocyst. Implementing both drainage methods (through the wall of the gastrointestinal tract and through the major duodenal papilla) in patients with pancreatic pseudocysts increases the condition’s duration and worsens the outcomes of endotherapy.

### 2.3. Passive Transpapillary Drainage during Transmural Drainage of Pancreatic Fluid Collections

In patients undergoing transmural drainage of walled-off pancreatic necrosis, the use of passive transpapillary drainage improves the results of endotherapy. Conversely, in patients undergoing transmural drainage of pancreatic pseudocysts, passive transpapillary drainage worsens the short- and long-term outcomes of endoscopic treatment and increases the duration of endotherapy.

The use of passive transpapillary drainage through pancreatic duct stenting should depend on the type of pancreatic fluid collection being treated. Endoscopic retrograde pancreatography should be performed to assess the integrity of the main pancreatic duct and to implement endotherapy in case a disruption of the pancreatic duct is identified for all patients undergoing endoscopic drainage of walled-off pancreatic necrosis, as opposed to patients with pancreatic pseudocysts. The endoscopic treatment of main pancreatic duct disruption constitutes the key component of endotherapy for walled-off pancreatic necrosis. Stenting of the main pancreatic duct in patients with main pancreatic duct disruption in the course of acute necrotizing pancreatitis increases the effectiveness of endotherapy for walled-off pancreatic necrosis, improves the long-term results of endoscopic treatment, and reduces the number of recurrent pancreatic fluid collections. The implication of the hypothesis for clinical practice is the implementation of endoscopic retrograde pancreatography in all patients with pancreatic necrosis to assess the integrity of the pancreatic duct and pancreatic duct stenting if a duct disruption is identified. In patients with pancreatic pseudocysts, stenting of the main pancreatic duct adversely affects the results of endoscopic treatment and prolongs endotherapy.

The literature contains few publications regarding passive transpapillary drainage in patients with acute pancreatitis, and the available data are contradictory. Endoscopic treatment of main pancreatic duct disruption in the course of acute necrotizing pancreatitis is one of the key elements of endotherapy for walled-off pancreatic necrosis. Stenting of the main pancreatic duct is an effective method for treating disruption of the pancreatic duct, and it improves the results of endoscopic treatment in patients with walled-off pancreatic necrosis [10]. Similar conclusions were presented by Trevino et al. [16], where the authors demonstrated that stenting of the main pancreatic duct during the transmural endoscopic drainage of fluid collections increased the effectiveness of endotherapy.

Passive transpapillary drainage performed during pancreatic pseudocyst drainage, reduces the effectiveness of endotherapy and prolongs endoscopic treatment. Hookey et al. found no significant differences in the effectiveness of treatment between patients undergoing only endoscopic ultrasound-guided transmural drainage versus the group where main pancreatic duct stenting (combined drainage) was applied in addition to transmural drainage [14]. They also reported a higher recurrence rate of fluid collections in the group treated with the combined approach compared to patients that underwent transmural drainage alone. Thus, they hypothesized that the main pancreatic duct impedes patency and maturation of the wall, thereby inhibiting the regression of pancreatic fluid collections [14]. Yang et al. [15] corroborated the Hookey et al. hypothesis [14]. They demonstrated that restoring the physiological outflow of pancreatic juices to the duodenum by stenting the pancreatic duct did not adversely affect the results of treatment in patients who underwent transmural drainage of pancreatic pseudocysts but negatively affected long-term outcomes of endoscopic treatment for pancreatic fluid collections [15].

In the cited papers [14,15,16], the majority of subjects were patients with pancreatic pseudocysts. The number of patients with walled-off pancreatic necrosis accounted for 0% to 20% of the study group depending on the publication [14,15,16]. Authors did not differentiate between the results of endoscopic treatment according to the type of collection [14,15,16], which made it difficult to compare the results of the above-mentioned studies in the context of the hypothesis presented in this publication, substantiating the need for further research on this subject.

Disruption of the main pancreatic duct occurs in most patients with acute pancreatitis. Partial pancreatic duct disruption is more common than complete disruption [6,10,17]. In a study that only included patients with acute necrotizing pancreatitis, Jang et al. showed better treatment outcomes in patients with partial disruption of the duct than in patients diagnosed with complete disruption [17]. In the same study, a higher number of relapses of pancreatic fluid collections were found in patients with complete duct disruption compared to those with partial disruption. The same conclusions were presented by Shrode et al. [18], who demonstrated that stenting the main pancreatic duct during endotherapy for pancreatic fluid collections was effective only in patients with partial duct disruption, whereas in patients with complete duct disruption, insertion of a stent into the main pancreatic duct did not provide any therapeutic benefit [18]. Despite worse outcomes of endoscopic treatment, stenting of the main pancreatic duct should also be used in patients with complete duct disruption, but only in the course of acute necrotizing pancreatitis. Endoscopic treatment, such as pancreatic duct stenting, should be administered in all patients diagnosed with main pancreatic duct disruption in the course of acute necrotizing pancreatitis, regardless of the type of disruption.

In this paper, we demonstrated the effectiveness of passive transpapillary drainage in patients with walled-off pancreatic necrosis, as opposed to patients with pancreatic pseudocysts, where main pancreatic duct stenting worsens the outcomes of endotherapy. In combination with active wall drainage, passive transpapillary drainage is an effective, minimally invasive treatment method for walled-off pancreatic necrosis. In contrast, in patients with pancreatic pseudocysts, the combination of transmural drainage with passive transpapillary drainage does not offer any therapeutic benefit.

### 2.4. Additional Conservative Treatment

As described above, the aim of endoscopic treatment of main pancreatic duct disruptions is to ensure physiological drainage of pancreatic juice into the duodenum, which consequently decreases leakage of pancreatic juice into the lumen of the collection communicating with main pancreatic duct. Similar results should be achieved with the use of somatostatin and its analogues (e.g., octreotide), which reduce the pancreatic exocrine secretions [19,20]. The results of clinical trials using somatostatin and its analogues in treatment of acute pancreatitis are controversial and ambiguous [20,21,22]. Somatostatin and its analogues are effective in the treatment of post-operative pancreatic fistulas and internal pancreatic fistulas in the course of pancreatitis [23,24,25]. If we define main pancreatic duct disruption as internal pancreatic fistula, it seems that the use of somatostatin and its analogues should hasten the healing of disruption, especially in connection with passive transpapillary drainage. However, that type of management requires additional studies to evaluate its efficacy.

## 3. Practical Guidelines

Endoscopic treatment of pancreatic duct disruption includes endoscopic sphincterotomy and implantation of a prosthesis into the main pancreatic duct. Passive transpapillary drainage (stenting of the pancreatic duct) decreases the pancreatic ductal pressure and helps to rapidly close the disruption. In the case of pancreatic pseudocysts communicating with the main pancreatic duct, passive transpapillary drainage aims to evacuate serous content from the collections to achieve complete regression of the pseudocysts. Patients with pancreatic pseudocysts and pancreatic duct disruption should undergo transpapillary drainage or transmural drainage of the pseudocysts. Implementing both drainage methods (transpapillary and transmural) in patients with pancreatic pseudocysts worsens the outcomes of endotherapy and increases its duration. The case is different in patients with walled-off pancreatic necrosis. The main reason for passive transpapillary drainage during transmural drainage of pancreatic necrosis is not to drain the necrotic content through the stent but to ensure physiological drainage of pancreatic juices into the duodenum. It helps to heal pancreatic duct disruption and decreases leakage of pancreatic juice into the lumen of collections, which consequently also helps to heal the collections. In all patients with walled-off pancreatic necrosis treated at our medical center, we attempted to perform endoscopic retrograde pancreatography to assess the morphology and integrity of the main pancreatic duct and the possible use of endoscopic treatment. In the case of the disruption of the pancreatic duct, endoscopic sphincterotomy was performed and pancreatic endoprosthesis was introduced into the main pancreatic duct, which was then replaced on average at 3, 6, 12, and 24 months or until there was no contrast flow outside the duct. The size and length of the pancreatic stent should be selected individually according to the fluoroscopic image of the main pancreatic duct during endoscopic retrograde pancreatography. The most common stents are straight or one-pigtail, the size of 5 or 7Fr, and the length of 9 or 12 cm. The scheme of passive transpapillary drainage in patients with pancreatic pseudocysts looks similar. The only difference is that larger pancreatic stents (8, 5, or 10Fr) are used more frequently.

## 4. Conclusions

In our opinion, not every patient during endoscopic treatment of pancreatic fluid collections requires stenting of the main pancreatic duct (endoscopic passive transpapillary drainage). The use of passive transpapillary drainage during endoscopic transmural drainage should depend on the type of pancreatic fluid collection. There is a need to perform endoscopic retrograde pancreatography with pancreatic duct stenting in all patients with pancreatic duct disruption during endotherapy for walled-off pancreatic necrosis which helps to improve the results of endoscopic treatment in patients with acute necrotizing pancreatitis. Conversely, passive transpapillary drainage should not be performed during transmural drainage in patients with pancreatic pseudocysts, even in cases of confirmed main pancreatic disruption as it negatively affects treatment outcomes. We emphasize, however, that further randomized trials are necessary to confirm the conclusions.

However, it is our opinion that, relying on the current state of knowledge, passive transpapillary drainage is an effective method for treating main pancreatic duct disruption in the course of acute necrotizing pancreatitis, and it improves the results of endoscopic treatment in patients with walled-off pancreatic necrosis. In patients with pancreatic pseudocysts, passive transpapillary drainage worsens the results of endoscopic treatment and should not be used even in cases of main pancreatic duct disruption.

## Figures and Tables

**Figure 1 jcm-10-00761-f001:**
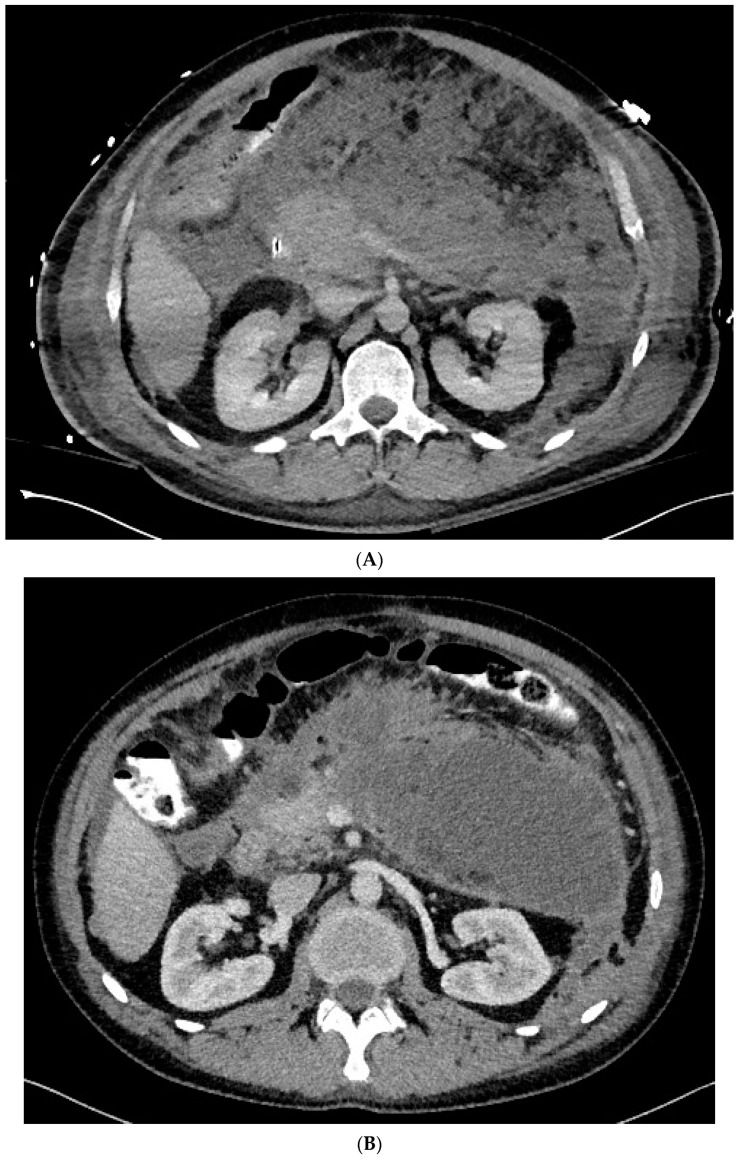
(**A**–**L**) Endoscopic treatment of walled-off pancreatic necrosis. In the second week of acute necrotizing pancreatitis, the acute necrotic collection (**A**) is visible in the abdominal contrast-enhanced computed tomography (CECT), which evolved in the sixth week of the illness duration into the symptomatic walled-off pancreatic necrosis (**B**). Patient qualified for endoscopic treatment (**C**–**F**) transmural drainage using the self-expanding metal stent (**C**,**D**) and endoscopic necrosectomy (**E**,**F**) was performed. In the second week of endotherapy, the endoscopic retrograde pancreatography (**G**–**J**) was performed. During pancreatography, the complete pancreatic duct disruption was stated (**G**–**I**) and transpapillary drainage was carried out (**J**). After achieving the treatment’s success and the complete regression of the necrotic collection, the transpapillary stent was observed in the bottom of the collection via the endoscopic view from the stomach’s side through the transmural stent (**K**). Control CECT confirmed the total regression of the collection (**L**).

**Figure 2 jcm-10-00761-f002:**
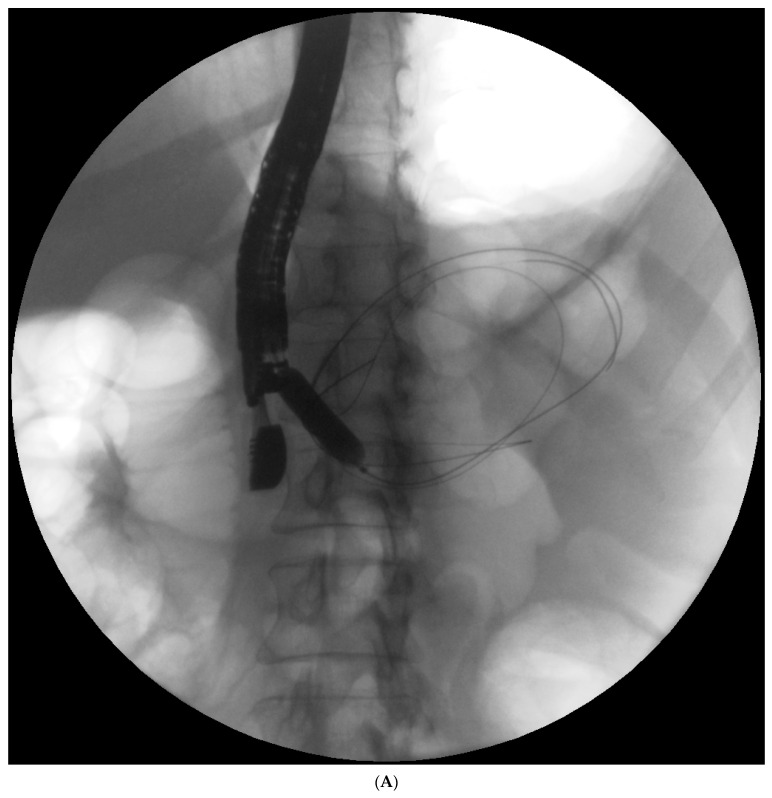
(**A**–**C**). Endoscopic treatment of the pancreatic pseudocyst. Transmural drainage using plastic stents was performed (**A**,**B**). The endoscopic retrograde pancreatography (**C**) was performed. During the pancreatography, complete disruption of the pancreatic duct was stated (**C**). Transpapillary access to the pseudocyst was achieved (**C**). Transmural access to the same collection was stated (**A**,**B**).

## Data Availability

Not applicable.

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
