# Peer review of "The Role of Endoscopic Transpapillary Stenting of the Main Pancreatic Duct during the Endoscopic Treatment of Pancreatic Fluid Collections"

_jcm, 2021, doi:10.3390/jcm10040761_

Round 1

Reviewer 1 Report

Please stretch out, that early ERCP is not recommended in acute pancreatitis by Guidelines but that your study deals with a special situation in severe acute pancreatitis.

Please give further information about type an size and length of used stents.

A reference to the amount of juice secretion in acute pancreatitis would be helpful.

Author Response

Reviewer #1

Please stretch out, that early ERCP is not recommended in acute pancreatitis by Guidelines but that your study deals with a special situation in severe acute pancreatitis.

Please give further information about type an size and length of used stents.

A reference to the amount of juice secretion in acute pancreatitis would be helpful.

Response:

Dear Reviewer,

In the beginning, I would like to thank you for a very positive revision of our manuscript.

We agree with the Reviewer, that early ERCP in acute pancreatitis is not recommended according to the guidelines. However, our publications concerns the late phase of acute pancreatitis complicated by pancreatic fluid collections, where ERCP should be considered. As we emphasize in our publication, currently, no guidelines clearly define the role of pancreatic duct stenting in patients undergoing the endoscopic drainage of pancreatic fluid collections. Moreover, the use of endoscopic techniques in the treatment of main pancreatic duct disruption due to acute pancreatitis has not been clarified. Performation of ERCP during the endoscopic treatment of pancreatic fluid collections is still controversial and mentions of it in literature are often contrary. A small number of publications is available in the current literature regarding ERCP in patients during late phase of acute pancreatitis, and the available data is often contradictory. In this paper, a comprehensive review of current literature was performed to identify publications on the role of pancreatic duct stenting in patients undergoing the endoscopic drainage of pancreatic fluid collections as consequences of acute pancreatitis. In this publication, an attempt was made to clarify this most controversial aspect of endotherapy for pancreatic fluid collections based on existing knowledge regarding the endoscopic treatment of acute pancreatitis sequelae. For above reasons, we think our paper is important and valuable. We believe our hypothesis of our paper is crucial in the clinical setting and it can help improve the results of pancreatic endotherapy for fluid collections resulting from acute pancreatitis.

According to Reviewer’s suggestion, we have added information about types and sizes of pancreatic stents. The size and length of the pancreatic stent should be selected individually according to fluoroscopic image of main pancreatic duct during endoscopic retrograde pancreatography. The most common stents are: straight or “one-pigtail”, the size of 5 or 7Fr and the length of 9 or 12 cm.

We fully agree with the Reviewer and according to Reviewer’s suggestion, we have added paragraph with information about juice secretion in acute pancreatitis. We agree with the Reviewer, that it is very helpful. In the mentioned paragraph, we included information about physiological secretion of pancreatic juice and pathological secretion of pancreatic juice in acute pancreatitis. We also included information about possibility of conservative treatment of pancreatic duct disruptions with use of somatostatin and its analogues, which reduce the pancreatic exocrine secretion.

We hope that made corrections will satisfy both the Reviewers and the Editors.

We hope that our corrections will make the manuscript meet the requirements for publication in “Journal of Clinical Medicine”.

With kind regards,

Mateusz Jagielski and Marek Jackowski

Reviewer 2 Report

I have evaluated the manuscript by Jagielski et al, which is interesting and very well written. 

I have only few suggestions:

  • I think it would be useful for the reader that the paragraph "clinical and research consequences" is divided in 2 sub-paragraph: one about the walled-off pancreatic stenting and one about pancreatic pseudocyst. 
  • at the bottom of page 4, I think there is a mistake "in the second of acute necrotizing pancreatitis". Did you want to write "second week"?
  • page 2: "some patients exhibit clinical symptoms". Which symptoms? Add them, please. 

Author Response

Reviewer #2

I have evaluated the manuscript by Jagielski et al, which is interesting and very well written. 

I have only few suggestions:

I think it would be useful for the reader that the paragraph "clinical and research consequences" is divided in 2 sub-paragraph: one about the walled-off pancreatic stenting and one about pancreatic pseudocyst. 

at the bottom of page 4, I think there is a mistake "in the second of acute necrotizing pancreatitis". Did you want to write "second week"?

page 2: "some patients exhibit clinical symptoms". Which symptoms? Add them, please. 

Response:

Dear Reviewer,

First of all, I would like to thank you very much for a very positive revision of our manuscript.

According to Reviewers’ suggestion, we devided the paragraph "clinical and research consequences" in sub-paragraphs: walled-off pancreatic necrosis and pancreatic pseudocyst. We corrected the text of our paper to make it more clear and readable.

Yes, we agree with Reviewer. It was our mistake. It should be: “in the second week of acute necrotizing pancreatitis”. We corrected our mistake.

It is information about indications for the interventional treatment of pancreatic fluid collections. The patients with persistent and symptomatic pancreatic fluid collections (clinical symptoms associated with the presence of pancreatic fluid collections) require interventional treatment. The main indications for interventional treatment of consequences of acute pancreatitis are infected pancreatic and peripancreatic fluid collections. Interventional treatment is also required in patients with clinical symptoms directly associated with the collections, such as compression symptoms (mechanical jaundice, ileus, etc.). Patients with asymptomatic pancreatic fluid collections, regardless of the size, do not require interventions. This information were added to the introduction section.

According to Reviewer’s recommendation, we reviewed the article for minor style and spell errors.

We hope that made corrections will satisfy both the Reviewers and the Editors.

We hope that our corrections will make the manuscript meet the requirements for publication in “Journal of Clinical Medicine”.

With kind regards,

Mateusz Jagielski and Marek Jackowski

Reviewer 3 Report

This paper reviews the evidence of transpapillary stenting in the context of acute pancreatic collections. I agree with the importance of radiological classification of these lesions. The main conclusion of this paper is that every patient with walled off necrotic  (WON) collections should undergo an ERCP and pancreatic stent, whilst this does not infer a favourable outcome in the context of a pseudocyst. The evidence presented for the use of a pancreatic stent in WON is from the author's own paper from 2018. This was a retrospective case series and I don't think a blanket recommendation on ERCP for all of these patients can be proposed based on this non-randomised series. Furthermore, the authors do not propose the timing of this intervention in the course of the patient's admission with pancreatitis, the type or size of the stents, or the duration of the pancreatic duct stenting. A hypothesis of why a small small calibre plastic stent can help drain a WON lesion filled with tenacious liquid and solid material (or augment transmural drainage) is not offered. It is the experience in our Centre that we pretty much never undertake an ERCP in the context of all of a WON collection, not least as anatomically this is often very challenging (distorted anatomy, oedematous duodenum exception). Our outcomes are good. I would be interested to hear of the 226 patients in their retrospective case series was an intention-to-treat, and the number of unsuccessful attempt to gain pancreatogram or stent pancreas this includes. There are further inconsistencies in the paper. In the introduction they discuss the fact that most acute necrotic collections regress spontaneously without intervention (second paragraph) yet in the conclusions they clearly advocate ERCP for all cases unless I am mistaken. I think this paper would benefit from a significant rewrite, with a systematic review of the literature on this topic including methodology to arrive at any cited papers. I think they should be welcome to quote their retrospective case series (reference 10) but place much less emphasis on such solid conclusions drawn from it, and rather speculate on the mechanisms and requirements for future study in the context of a randomised series.

Author Response

Reviewer #3

This paper reviews the evidence of transpapillary stenting in the context of acute pancreatic collections. I agree with the importance of radiological classification of these lesions. The main conclusion of this paper is that every patient with walled off necrotic  (WON) collections should undergo an ERCP and pancreatic stent, whilst this does not infer a favourable outcome in the context of a pseudocyst. The evidence presented for the use of a pancreatic stent in WON is from the author's own paper from 2018. This was a retrospective case series and I don't think a blanket recommendation on ERCP for all of these patients can be proposed based on this non-randomised series. Furthermore, the authors do not propose the timing of this intervention in the course of the patient's admission with pancreatitis, the type or size of the stents, or the duration of the pancreatic duct stenting. A hypothesis of why a small small calibre plastic stent can help drain a WON lesion filled with tenacious liquid and solid material (or augment transmural drainage) is not offered. It is the experience in our Centre that we pretty much never undertake an ERCP in the context of all of a WON collection, not least as anatomically this is often very challenging (distorted anatomy, oedematous duodenum exception). Our outcomes are good. I would be interested to hear of the 226 patients in their retrospective case series was an intention-to-treat, and the number of unsuccessful attempt to gain pancreatogram or stent pancreas this includes. There are further inconsistencies in the paper. In the introduction they discuss the fact that most acute necrotic collections regress spontaneously without intervention (second paragraph) yet in the conclusions they clearly advocate ERCP for all cases unless I am mistaken. I think this paper would benefit from a significant rewrite, with a systematic review of the literature on this topic including methodology to arrive at any cited papers. I think they should be welcome to quote their retrospective case series (reference 10) but place much less emphasis on such solid conclusions drawn from it, and rather speculate on the mechanisms and requirements for future study in the context of a randomised series.

Response:

Dear Reviewer,

First of all, I would like to thank you for positive review of our manuscript. According to Reviewer’s suggestion, we have made the appropriate corrections in the manuscript. In the further paragraphs of this response we would like to address each every sentence of the review and thoroughly respond to Reviewer’s suggestion.

Reviewer:

This paper reviews the evidence of transpapillary stenting in the context of acute pancreatic collections. I agree with the importance of radiological classification of these lesions. The main conclusion of this paper is that every patient with walled off necrotic  (WON) collections should undergo an ERCP and pancreatic stent, whilst this does not infer a favourable outcome in the context of a pseudocyst. The evidence presented for the use of a pancreatic stent in WON is from the author's own paper from 2018. This was a retrospective case series and I don't think a blanket recommendation on ERCP for all of these patients can be proposed based on this non-randomised series.”

Response:

Our review paper is in fact a description of our observations in context of data available in the current literature. In our medical center we treat c.120 patients with pancreatic fluid collections a year. We are currently running a randomized trial concerning benefits of transpapillary drainage during endotherapy of pancreatic fluid collections (pancreatic pseudocysts and walled-off pancreatic necrosis). Preliminary conclusions from our randomized trial were presented in this review paper. According to our observations, in patients undergoing transmural drainage of walled-off pancreatic necrosis, the use of passive transpapillary drainage improves the results of endotherapy. Conversely, in patients undergoing transmural drainage of a pancreatic pseudocyst, passive transpapillary drainage worsens the short-term and long-term outcomes of endoscopic treatment and increases the duration of endotherapy. Moreover, our observations show, that patients with pancreatic pseudocysts and pancreatic duct disruption should whether undergo transpapillary drainage of transmural drainage of pseudocyst. Implementing both drainage methods (through the wall of gastrointestinal tract and through the major duodenal papilla) in patients with pancreatic pseudocysts worsens the outcomes of endotherapy and increases its duration. In current review paper we wanted to emphasize that observation, which will be proven in further original papers from our medical center. We agree with the Reviewer, that our retrospective paper from 2018 may not serve as guidelines. We also see the need for further publications describing randomized trials connected with this issue. That is why we run such trials in our medical center.

Reviewer:

Furthermore, the authors do not propose the timing of this intervention in the course of the patient's admission with pancreatitis, the type or size of the stents, or the duration of the pancreatic duct stenting. A hypothesis of why a small small calibre plastic stent can help drain a WON lesion filled with tenacious liquid and solid material (or augment transmural drainage) is not offered.”

Response:

According to the Reviewer’s suggestion, we added data about timing of intervention, type and size of the pancreatic stent and also duration of transpapillary drainage. The basis of the treatment is to perform transpapillary drainage during ongoing transmural drainage of walled-off pancreatic necrosis- never in reverse. Transpapillary drainage performed before transmural drainage leads to secondary infection of the necrotic collection. Consequently, it worsens the clinical condition of patient due to inability to perform complete drainage transpapillary route. That is why active transmural drainage of pancreatic necrosis with endoscopic necrosectomy is performed in the first place, and only during ongoing transmural drainage we perform ERCP with pancreatic duct stenting. Despite our observations in reference to data from the literature, we still think, that the decision as to when to perform pancreatography remains a highly controversial topic.

In our paper we presented endotherapy of pancreatic fluid collection in late phase of acute pancreatitis (more than four weeks since the admission). The main indications for interventional treatment of consequences of acute pancreatitis are infected pancreatic and peripancreatic fluid collections. Interventional treatment is also required in patients with clinical symptoms directly associated with the collections, such as compression symptoms (mechanical jaundice, ileus, etc.). Patients with asymptomatic pancreatic fluid collections, regardless of the size, do not require interventions.

The main reason for passive transpapillary drainage (endoscopic sphincterotomy and implantation of a prosthesis into the main pancreatic duct) is not to drain the necrotic content through the stent, but to ensure physiological drainage of pancreatic juice into the duodenum. It helps to heal pancreatic duct disruption and decreases leakage of pancreatic juice into the lumen of collections, which consequently also helps to heal the collection.

Reviewer:

It is the experience in our Centre that we pretty much never undertake an ERCP in the context of all of a WON collection, not least as anatomically this is often very challenging (distorted anatomy, oedematous duodenum exception). Our outcomes are good. I would be interested to hear of the 226 patients in their retrospective case series was an intention-to-treat, and the number of unsuccessful attempt to gain pancreatogram or stent pancreas this includes.”

Response:

Yes, we agree that ERP (endoscopic retrograde pancreatography) in patients with pancreatic fluid collections is often very challenging. In our retrospective study from 2018, we  attempted to perform ERP in all patients with pancreatic necrosis in order to assess the integrity of the main pancreatic duct and performed endotherapy in case of pancreatic duct disruption (intention-to-treat). ERP was performed in 204/226 (90.27%) patients. Eighty five patients underwent ERP during the first endotherapy procedure for WOPN (walled-off pancreatic necrosis). The remaining 119 patients underwent pancreatography within 19 (SD=24.79) [range 2–145] days from the beginning of endotherapy. In 176 patients, ERP was performed on the first attempt. In 19 patients, it was successful on the second attempt, in 7 patients on the third attempt, and in 2 patients on the fourth attempt. No more than four attempts at ERP were made. In 22/226 (9.73%) patients, attempts to perform ERP were unsuccessful—12 patients did not complete endoscopic therapy, and in 10 patients, failed MPD (main pancreatic duct) cannulation was caused by duodenal deformation and impression preventing the introduction of the device (luminal compression; 8 patients), or it was impossible to identify the major duodenal papilla in the duodenal infiltration (2 patients). In the group of 22 patients who did not undergo ERP, the outcomes of endoscopic treatment were significantly worse compared to the group of 204 patients who underwent ERP—there was a lower rate of therapeutic success of WOPN endotherapy (81.82 vs. 95.1%, p 0.03), a higher rate of recurrent PFCs (50 vs. 9.31%, p<0.01), and a lower rate of long-term success of WOPN treatment (20 vs. 80.2%, p<0.01). ERP was performed in 204/226 (90.27%) patients with WOPN. In 166/204 (81.37%) patients, there was contrast flow outside the MPD (MPD disruption). Partial and complete disruption of the MPD were identified in 103 (50.49%) and 63 (30.89%) out of 204 patients, respectively. In 17/204 (8.33%) patients, only a fragment of the pancreatic duct filled with contrast, without contrast flow outside the duct. In 21/204 (10.29%) patients, the MPD was normal on ERP. Based on the results of additional imaging diagnostics, 52 patients were diagnosed with DDS (disconnected duct syndrome) (28 patients with complete MPD disruption, 8 patients with only a fragment of the MPD filled with contrast, without spill, and 16 patients in whom it was not possible to perform ERP). Endoscopic treatment was used in all 166 patients with MPD disruption. In 12 patients, the endoprosthesis could not be introduced into the MPD, only endoscopic sphincetrotomy. was performed (3 patients with partial MPD disruption, 9 with complete MPD disruption). In the remaining 154 patients, endoprosthesis was introduced into the MPD. In patients with partial MPD disruption, type 1 intervention was used in 15 patients, and type 3 prosthesis implantation procedure was performed in 85 patients. In the group of patients with complete MPD disruption, type 1 intervention was performed in 19 patients, type 2—in 13 patients, and 22 patients underwent type 3 prosthesis implantation. Repeated ERP was performed in 161/166 (96.99%) patients with MPD disruption identified during the first procedure. Five patients did not complete endoscopic treatment; these patients have not returned for scheduled hospitalizations to replace their prostheses. The average duration of prosthesis use was 341.9 (SD = 251.8) [range 33–730] days; mean number of endoscopic procedures with MPD prosthesis was 2.34 (SD = 1.15) [range 1–5]. The success of endoscopic treatment of MPD disruption was achieved in 138/161 (85.71%) patients with WOPN. Twelve patients are still undergoing endoscopic treatment for MPD disruption. Eleven patients with MPD disruption required surgical treatment due to the recurrence of PFCs.

Reviewer:

There are further inconsistencies in the paper. In the introduction they discuss the fact that most acute necrotic collections regress spontaneously without intervention (second paragraph) yet in the conclusions they clearly advocate ERCP for all cases unless I am mistaken.”

Response:

In the introduction section we shortly described the natural history of the course of pancreatic fluid collections as the consequences of acute pancreatitis. In further paragraphs of our review paper, we emphasize the indications for interventional treatment of pancreatic fluid collections with special emphasis on endotherapy. The patients with persistent and symptomatic pancreatic fluid collections require interventional treatment. Patients with asymptomatic pancreatic fluid collections, regardless of the size, do not require interventions. Therefore conclusions about ERCP with stenting of pancreatic duct are valid only for patients included to interventional treatment. Therefore our conclusions are valid for patients with persistent and symptomatic pancreatic fluid collections during transmural drainage. We corrected the text of our paper to make it more clear and readable.

Reviewer:

“I think this paper would benefit from a significant rewrite, with a systematic review of the literature on this topic including methodology to arrive at any cited papers. I think they should be welcome to quote their retrospective case series (reference 10) but place much less emphasis on such solid conclusions drawn from it, and rather speculate on the mechanisms and requirements for future study in the context of a randomised series.

Response:

According to Reviewers’ suggestions the content of our paper was corrected according to our observations and other studies from our center, so to emphasize our assumptions and show our conclusions in clear and readable way. We agree with the Reviewer, that the systematic review of the literature would have the greatest substantive value, which attempted in our paper. We rewritten our paper to emphasize our observations based on data from the literature. We corrected the article to place much less emphasis on solid conclusions drawn from it, and rather speculate on the mechanisms and requirements for future study in the context of a randomized series. We hope our corrections will be satisfactory for the Reviewer.

Moreover, we reviewed the article for minor style and spell errors.

In conclusion, we hope that made corrections will satisfy both the Reviewers and the Editors.

We hope that our corrections will make the manuscript meet the requirements for publication in “Journal of Clinical Medicine”.

With kind regards,

Mateusz Jagielski and Marek Jackowski

Round 2

Reviewer 3 Report

This paper reviews the evidence of trans-papillary stenting in the context of acute pancreatic collections. I agree with the importance of radiological classification of these lesions. The main conclusion of this paper is that every patient with walled off necrotic collection should undergo an ERCP and pancreatic stent, whilst this does not infer a favourable outcome in the context of a pseudocyst. The evidence presented for the use of a pancreatic stent in walled off necrosis is from the author's own paper from 2018. This was a retrospective case series and I don't think a blanket recommendation on ERCP for all of these patients can be proposed based on this single paper. Furthermore, the authors do not propose the timing of this intervention in the course of the patient's admission with pancreatitis; the type or size of the stents; or the duration of the pancreatic duct stenting. It is the experience in our Centre that we pretty much never undertake an ERCP in the context of walled off necrotic collections, not least as anatomically this is often very challenging (distorted anatomy, oedematous duodenum). I would be interested to hear of the 226 patients in their retrospective case series was an intention-to-treat, and the number of unsuccessful attempts to permit PD stenting or pancreatogram this includes. There are further inconsistencies in the paper. In the introduction they discuss the fact that most acute necrotic collections regress spontaneously without intervention (second paragraph) yet in the conclusions they clearly advocate ERCP for all cases unless I am mistaken. I think this paper would benefit from a significant rewrite, with a systematic review of the literature on this topic including methodology to arrive at any cited papers. I think they should be welcome to quote their retrospective case series (reference 10) but place much less emphasis on such solid conclusions drawn from it, and rather speculate on the mechanisms and requirements for future study in the context of a randomised series.  

Author Response

(The authors gave the same response as above.)

Round 3

Reviewer 3 Report

Thank you for your changes, I think the paper has been significantly enhanced.